# A Unified, Scalable Framework for
# Neural Population Decoding

**Mehdi Azabou**[1,*]**, Vinam Arora**[1]**, Venkataramana Ganesh**[1]**, Ximeng Mao**[2,3] **,**
**Santosh Nachimuthu**[1]**, Michael J. Mendelson**[1]**, Blake Richards**[2,4]**,**
**Matthew G. Perich**[2,3]**, Guillaume Lajoie**[2,3]**, Eva L. Dyer**[1,*]
[1] Georgia Tech, [2] Mila, [3] Université de Montréal, [4] McGill University

## Abstract

Our ability to use deep learning approaches to decipher neural activity would likely benefit from greater scale, in terms of both model size and datasets. However, the integration of many neural recordings into one unified model is challenging, as each recording contains the activity of different neurons from different individual animals. In this paper, we introduce a training framework and architecture designed to model the population dynamics of neural activity across diverse, large-scale neural recordings. Our method first tokenizes individual spikes within the dataset to build an efficient representation of neural events that captures the fine temporal structure of neural activity. We then employ cross-attention and a PerceiverIO backbone to further construct a latent tokenization of neural population activities. Utilizing this architecture and training framework, we construct a large-scale multi-session model trained on large datasets from seven nonhuman primates, spanning over 158 different sessions of recording from over 27,373 neural units and over 100 hours of recordings. In a number of different tasks, we demonstrate that our pretrained model can be rapidly adapted to new, unseen sessions with unspecified neuron correspondence, enabling few-shot performance with minimal labels. This work presents a powerful new approach for building deep learning tools to analyze neural data and stakes out a clear path to training at scale.

## 1  Introduction

Recent advances in machine learning, particularly in the context of large-scale pretrained models like GPT [1, 2, 3, 4], have showcased the immense potential of scaling up, both the terms of the size of datasets and models [5, 6]. Similarly, in neuroscience, there is a growing need for a foundational model that can bridge the gaps between diverse datasets, experiments, and individuals, allowing for a more holistic understanding of brain function and information processing [7]. The development of such a model would allow researchers to uncover underlying patterns and interactions within neural populations [8], and potentially allow for more robust decoding of brain states.

However, creating a large-scale neural decoding model that can effectively combine spiking datasets from various sources is a complex challenge [9]. One of the central challenges is the lack of a shared "vocabulary" in neural recordings. Unlike the case for text—wherein every document written in a given language shares a basic lexicon for tokenization—there is no one-to-one correspondence between neurons in different individuals. As such, every recording from a different individual involves a unique set of neurons that cannot be easily aligned with another set. An additional core challenge lies in the inherent variability of neuron sets observed across different days [10]. Even when monitoring the same individual, variability in electrode/tissue interface can lead to distinct

---

*Contact: {mazabou,evadyer}@gatech.edu. Project page and code: https://poyo-brain.github.io

neuron sets which, despite advanced sorting methods, can lead to inconsistencies in input channels across sessions [11, 12, 9]. Overall, this lack of correspondence across recordings complicates the integration of information from different experiments and individuals, ultimately hampering efforts to construct a unified perspective on population-level interactions and dynamics in the brain.

In response to these challenges, we propose a new framework for large-scale training on neural spiking data called POYO (**P**re-training **O**n man**Y** neur**O**ns).[2] This framework is designed to enable scalable and efficient training across multiple sessions of neural recordings, even when spanning different sets of neurons with no known correspondence. Our approach centers around a novel tokenization scheme that transforms individual neural action potentials, or "spikes", into discrete tokens, thereby preserving the neural code's finest temporal structure while simultaneously enhancing computational efficiency. The resulting tokenization not only allows for a more effective representation of neural activity but also paves the way for training on larger volumes of data. We combine this input tokenization method with an architecture that builds on the PerceiverIO [13] to compress the input spikes into a latent space and learns interactions across spikes in time and across neurons.

We evaluate the performance of our proposed approach on data from over 158 sessions from open electrophysiology datasets from seven non-human primates (NHPs), spanning over 27,373 units and 100 hours of recordings. We demonstrate that through pretraining on large amounts of data, we can transfer with very few samples (few-shot learning) and thus improve overall brain decoding performance. Our work not only presents an innovative framework for training large models on neuroscience datasets, but also offers insights into the scaling laws that govern decoding from neural populations. By enabling the development of large pretrained models for neural decoding, our approach advances the field of brain-machine interfaces and other decoding applications.

The main contributions of this work include:

- *A framework for large-scale training on neural recordings:* We present a novel framework for training transformer models end-to-end on multi-session and across-individual electrophysiology datasets derived from neural populations, enabling efficient and effective decoding from a diverse range of neural recordings.

- *Innovative spike-based tokenization strategies:* We introduce a fundamentally different way to tokenize neural population activity. Our approach tokenizes individual spikes (events) across neural populations, preserving fine temporal structure and enhancing computational efficiency by adopting a sparse representation of the data.

- *Pre-trained models for neural decoding:* We build two large pretrained models (POYO-1, POYO-mp) that can be fine-tuned on new sessions and across recordings from different animals and new behavioral tasks. We will make the weights and code available, and provide both pretrained models as a resource to the community.

## 2 Approach

The transformer architecture [14], originally introduced in the context of natural language processing (NLP), has shown remarkable flexibility and effectiveness in various domains, especially in the presence of large and diverse datasets [15, 3]. In this work, we explore how to leverage this versatility in the neural data domain.

### 2.1 Tokenizing neural population activity

Neurons communicate asynchronously using electrical impulses called spikes. The timing and frequency of spikes encode signals that convey information about the external world and coordinate the internal dialogue within the brain. In many neuroscience experiments, neural activity is recorded from the brain through multiple electrodes, and then processed [16] to extract the spiking events for a set of neural "units" [17].[3] The resulting data are multi-variate event-based sequences that are typically very sparse relative to the number of recorded time points.

---

[2]Poyo is the exclamation used by Kirby, who has an insatiable appetite. Similarly, POYO consumes spikes from many neural datasets and combines them into one unified model.

[3]We use the term units instead of neurons because recordings may include both single-units (isolated neurons) and multi-units (contain spikes from multiple neurons).

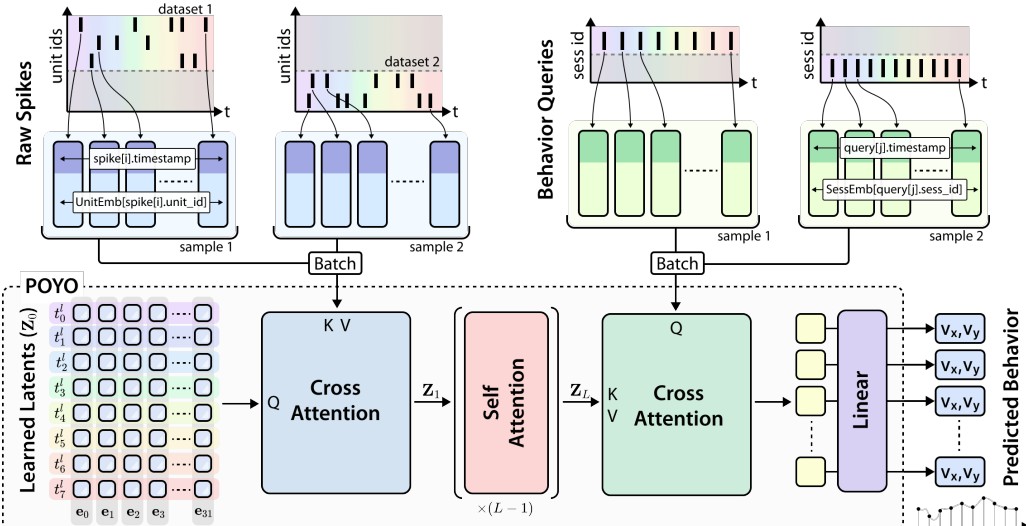

Figure 1: *Overview of our approach.* Input spike tokens are compressed into a smaller set of latent tokens which are processed through multiple self-attention blocks, finally time-varying outputs are predicted by querying the latent space. All tokens in this model are assigned a timestamp, which are used towards rotary position encoding.

Neurons and their spikes are of course not independent of one another. Rather, they are part of a complex, interwoven tapestry of neural activity, with each spike contributing to a collective dialogue across billions of neurons. The true challenge and opportunity lie in interpreting these spikes not in isolation, but in the context of this broader "conversation."

When trying to scale up and train on many sessions of data, we are faced with the challenge of having recordings of neural conversations with completely different set of "speakers" for which we don't know the identity or their functional tuning (or what they respond to). This is because each time we record from the brain, we are tuning into a conversation between a new set of speakers. However, as with language, there is some reason to think that neurons are ultimately communicating on similar topics, e.g. sensations from the external world, internal states of the body, muscle commands, etc. In other words, the lexicon may be different, but everyone is talking about similar subjects. Despite the variability of our data, our aim is to decipher these "conversations" in a way that generalizes to new neural datasets without fixed or known correspondence across their inputs.

**The neural tokenizer.** Based upon these motivations, we propose a novel approach for tokenizing neural population activity where *each spike is represented as a token* (see Figure 1). In this case, each token can be defined in terms of *which unit it came from* (via a learnable embedding) and the time that the spike event was detected. By representing our data in this way, we never define or fix the expected number of units, and the model can ingest populations of arbitrary size, and thus can be trained across many datasets. At the same time, this approach also avoids having to specify a specific time resolution for the neural code, as is the case with binning, and gets rid of the accumulation of a lot of sparse tokens containing no events.

More concretely, we assign each unit a unique identifier and a corresponding $D$-dimensional learnable embedding. Let $\mathrm{UnitEmbed}(\cdot)$ denote a lookup table that associates each unit to its unit embedding. Akin to word embeddings that encode semantic meaning and relationship between words, the unit embedding space encodes something about the "speaker's" identity and role in neural computation. A unit fires a sequence of spikes in a context window of time $[0, T]$. Each spike will be represented by a token characterized by $(\mathbf{x}_i, t_i)$, where

$$\mathbf{x}_i = \mathrm{UnitEmbed}(\text{spike } i'\text{s unit})$$

is the learned embedding associated with the unit that emitted the spike and $t_i$ is the event timestamp.

Collectively, and for an arbitrary set of units, we combine all the spikes into a sequence of length $M$. The neural population activity is represented by $[\mathbf{x}_1, \ldots, \mathbf{x}_M]$ and their corresponding times $[t_1, \ldots, t_M]$. While the size of our context window $T$ stays fixed, $M$ will vary depending on the number of units in the population and their firing rates (see Figure 1). Note that all spikes from a specific unit have the same unit embedding, and only differ in their timing.

## 2.2 Building the latent space

**Compressing the input sequence.** Rather than processing the input sequences with self-attention, which is quadratic in sequence length and can be expensive for long sequences, we use a perceiver encoder module [18] that summarizes the input sequence into a shortened "latent" sequence. Let's consider a sequence of $M$ input tokens $\mathbf{X} = [\mathbf{x}_1, \ldots, \mathbf{x}_M]$ and a sequence of learned latent tokens $\mathbf{Z}_0 = [\mathbf{z}_{0,1}, \ldots, \mathbf{z}_{0,N}]$, where $\mathbf{z}_{0,i} \in \mathbb{R}^D$ and $N \ll M$. We use cross-attention to pull information from the input sequence into a compressed latent sequence. The queries $\mathbf{Q} = \mathbf{W}_q \mathbf{Z}_0$ are a projection of the learned latent tokens $\mathbf{Z}_0$ and the keys and values are a projection of the input tokens: $\mathbf{K} = \mathbf{W}_k \mathbf{X}$ and $\mathbf{V} = \mathbf{W}_v \mathbf{X}$, respectively. The cross-attention operation can be expressed as follows:

$$\mathbf{Z}_1 \leftarrow \text{Cross-Attn}(\underset{\underset{\mathbf{Z}_0}{\uparrow}}{\mathbf{Q}}, \underset{\overset{\mathbf{X}}{\diagup\diagdown}}{\mathbf{K}}, \mathbf{V}) = \text{softmax}\left(\frac{\mathbf{Q}\mathbf{K}^T}{\sqrt{d_k}}\right)\mathbf{V}. \tag{1}$$

We use the standard transformer block with pre-normalization layers and feed-forward nets.

**Attention in the latent space.** Following the compression of the input sequence through cross-attention, we apply multiple self-attention blocks on the latent token sequence, which now costs $O(N^2)$ instead of $O(M^2)$ with $N \ll M$. Let $\mathbf{Z}_l$ denote the embedding of the latent tokens at the $l$th layer. The self-attention operation can be expressed as follows:

$$\mathbf{Z}_{l+1} \leftarrow \text{Self-Attn}(\underset{\underset{\mathbf{Z}_l}{\diagup\mid\diagdown}}{\mathbf{Q}, \mathbf{K}, \mathbf{V}}) = \text{softmax}\left(\frac{\mathbf{Q}\mathbf{K}^T}{\sqrt{d_k}}\right)\mathbf{V} \tag{2}$$

We denote the final latent sequence obtained after $L$ layers as $\mathbf{Z}_L = [\mathbf{z}_{L,1}, \ldots, \mathbf{z}_{L,N}]$.

## 2.3 Encoding relative timing information

To incorporate timing information, we leverage rotary position encoding (RoPE) [19] across all attention layers in the architecture. Unlike traditional position encoding methods that inject absolute position information into the token's embedding, RoPE applies a rotation operation in the query, key embedding space, allowing each token in a sequence to attend to others based on its relative positional information.

Recall that each input token $i$ has an associated timestamp $t_i$. We will also assign a timestamp to each of the latent tokens: we divide the latent tokens into groups of equal size and spread each group uniformly over the context window $[0, T]$. This method allows us to capture temporal relationships between the latent tokens and the input tokens, enabling a temporal understanding of the encoded sequence. By distributing the latent tokens evenly within the context window $[0, T]$, we create a structured temporal representation, which preserves and propagates temporal information all the way through the model. A detailed formulation of the relative encoding can be found in Appendix A.1.

## 2.4 Querying the latent space

Having built a latent space which can encode and model any population of units, we now want a flexible way to readout behavioral variables. The output of the latent encoder is the latent sequence of size $N$, while the desired output can be any arbitrary sequence of length $P$. Let us consider the task of hand velocity decoding for example, in the context window $[0, T]$, the length of the output sequence will depend on the sampling frequency. Since we aspire to train on datasets sourced from various labs, the sampling rate can differ significantly. We thus need a flexible mechanism for predicting outputs of varying lengths and querying from the neural activity at specific points in time.

We define a sequence of output tokens $\mathbf{Y}_0 = [\mathbf{y}_{0,1}, \cdots, \mathbf{y}_{0,P}]$, where $P$ is the number of output time points, which can change across sequences. Each output token is defined by $(\mathbf{y}_{0,i}, t_i^{out})$. Initially all the output tokens within a session are set to the same learned embedding $\mathbf{y}_{0,i} = \mathbf{y}_0 \in \mathbb{R}^D \; \forall i$. The output $\mathbf{Y}$ is obtained by querying the latent space through cross-attention:

$$\mathbf{Y} \leftarrow \text{Cross-Attn}(\underset{\underset{\mathbf{Y}_0}{\uparrow}}{\mathbf{Q}}, \underset{\overset{\mathbf{Z}_L}{\diagup\diagdown}}{\mathbf{K}}, \mathbf{V}) = \text{softmax}\left(\frac{\mathbf{Q}\mathbf{K}^T}{\sqrt{d_k}}\right)\mathbf{V} \tag{3}$$

Since we use rotatary embeddings, and both the latent and output tokens have assigned timestamps, the querying mechanism can leverage the relative position of latent and output tokens to extract the temporally relevant context that enable prediction of the behavioral variable of interest.

**Session embeddings.** To account for variability of the experimental setups in real world settings, we propose to define a learnable session embedding which captures the hidden experimental variables. This information is injected into the output query $\mathbf{y}_0$, where again we use a lookup table to register each new session we encounter. To illustrate how we probe the model to generate an output, we can think about it as asking a question of the form:

*"Predict [BEHAVIOR] at time [TIMESTAMP] under the experimental conditions of [SESSION ID]".*

This produces a set of output tokens of the dimension of the number of queries which we then pass through an MLP to generate the behavioral variables of the desired dimension (e.g., 2D for hand velocities in a planar movement task).

### 2.5 Unit identification in new sessions

Our design of the unit embedding space allows our model to learn latent information about the units it encounters, as well as capture the relationship between units in the population. Given a new recording with unidentified units, we can transfer our model by mapping these new units into the unit embedding space. To do this, we introduce an approach that we call "unit identification", which leverages gradient descent to learn the embeddings of new units. In this approach, we freeze all existing weights of the model and simply add new rows to the $\mathrm{UnitEmbed}(\cdot)$ lookup table for each of the new units, as well as a new session embedding. Notably, the bulk of our model which maps the neural population activity to behavior is unchanged and is simply transferred to the new dataset. In our experiments, we find that this approach is surprisingly effective and allows us to rapidly integrate new datasets into the same underlying model. Further details and analysis on the robustness of this approach are presented in Appendix C.2.

## 3 Experiments

In this section, we demonstrate the promise of our approach for large-scale training and examine the benefit of scaling in neural population decoding.

### 3.1 Datasets and experiment setup

One of the key advantages of our approach is its ability to scale to handle large amounts of neural data, including sessions from different numbers of neurons, across different tasks and recording setups, and from different animals. Thus we set out to build a diverse dataset large enough to test our approach. We curated a multi-lab dataset with electrophysiological recordings from motor cortical regions, where neural population activity has been extensively studied [10], and deep learning tools and benchmarks have recently been established [27]. In total, we aggregated 178 sessions worth of data, spanning 29,453 units from the primary motor (M1), premotor (PMd), and primary somatosensory (S1) regions in the cortex of 9 nonhuman primates (see Table 1). We place this in the context of standard analyses within a single lab or paper which typically involve 10's of sessions and a few hundred neurons.

All of these neural recordings were collected while the animals performed various motor tasks that vary in their inherent complexity (see Figure 2A-B). The center-out (CO) task is relatively stereotyped, with the animal making a reach to one of eight targets after receiving a go cue, and

| Study | Regions | # Indiv | # Sess | # Units | # In | # Out | Tasks |
|---|---|---|---|---|---|---|---|
| Perich et al. [20, 21, 22, 23] | M1, PMd | 4 | 117 | 11,557 | 143M | 20M | CO, RT |
| Churchland et al. [24] | M1 | 2 | 9 | 1,728 | 706M | 87M | CO |
| Makin et al. [25] | M1, S1 | 2 | 47 | 14,899 | 123M | 15M | RT |
| Flint et al. [26] | M1 | 1 | 5 | 957 | 7.9M | 0.3M | CO |
| NLB-Maze [27] | M1 | 1 | 1 | 182 | 3.6M | 6.8M | Maze |
| NLB-RTT [27] | M1 | 1 | 1 | 130 | 1.5M | 2.8M | RT |

Table 1: *Datasets used in this work.* CO: Center-Out, RT: Random Target.

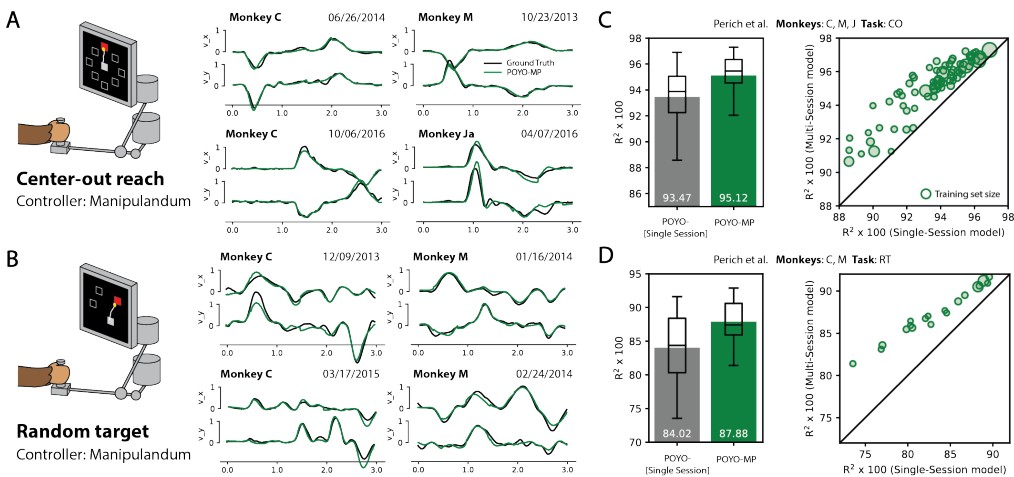

Figure 2: *Building a multi-session model spanning multiple animals and tasks.* (A) Center-out reach and (B) Random target task [20], along with examples of true and predicted behavior (x,y velocities). In (C-D), we show the decoding performance for single-session models (gray) and the `POYO-mp` multi-session model (green).

then returning to the center. In contrast, the random target (RT) task is significantly more complex. The animals make continuous and self-paced movements with new targets appearing in succession at random locations in the workspace. In addition to the greater exploration of the workspace and heterogeneity in movements, this task allows individual to plan their next movement while finishing the execution of the current movement leading to greater complexity in neural dynamics. Other sources of variability that we find across labs include the: choice of pre-processing algorithms (spike sorting or threshold crossing analysis), type of controller (manipulandum, touch screen), and sampling rate when recording behavior (100Hz to 1kHz). We do not re-process the data or attempt to standardize it across labs or tasks. Further details on the datasets are provided in Appendix B.

**Experiment setup.** Throughout all of our experiments, we use a context window of $1s$ and do not segment data into trials during training. We only use the trial structure when reporting the decoding performance, in particular, center-out sessions are evaluated during the reaching movement. We train the model with $N = 512$ latent tokens and a dimension $D = 128$. We use the LAMB optimizer [28], and employ a cosine decay of the learning rate at the end of training. For every session, we holdout $20\%$ of the trials for testing, and $10\%$ for validation. We use 1-GPU and 8-GPU setups for single-session and multi-session models, respectively. All details can be found in Appendix A.

### 3.2 Testing the model on single sessions

To first investigate the performance of our architecture when trained on a single session, we trained single-session models on 100 different recording sessions acquired from three nonhuman primates (Monkey C, Monkey M, Monkey Ja), each containing anywhere from 10 to 106 minutes of data [29]. In all of these recordings, the same behavioral task setup, behavioral recording apparatus (10 ms temporal resolution), and spike sorting procedure was used.

Across all 100 single-session models, we obtained an average R2 of 0.9347 on CO and 0.8402 for RT sessions. When we compared these single-session results with existing models for neural decoding, including a Wiener Filter, GRU and MLP [30], we found that our approach consistently outperforms these baselines, with even greater improvements observed on the RT task. Additionally, we found that our single-session models are stable for a wide range of hyperparameters (Appendix C.3).

### 3.3 `POYO-mp`: Building a large pretrained across-animal, multi-session model

To investigate the question of how training with more sessions of data can improve brain decoding, we trained a large model (`POYO-mp`, 24 layers) on all 100 sessions that we studied in our single-session analysis. In total, we used 9,789 units and 4,367 neuron-hours (number of neurons $\times$ amount of time recorded) to train our `POYO-mp` model.

| | Method | *Same animal, New day*
Monkey C - CO (2) | *New animal*
Monkey T - CO (6) | Monkey T - RT (6) |
|---|---|---|---|---|
| **From scratch** | Wiener Filter | $0.8860 \pm 0.0149$ | $0.6387 \pm 0.0283$ | $0.5922 \pm 0.0901$ |
| | GRU | $0.9308 \pm 0.0257$ | $0.8041 \pm 0.0232$ | $0.6967 \pm 0.1011$ |
| | MLP | $\underline{0.9498} \pm 0.0119$ | $\underline{0.8577} \pm 0.0242$ | $\underline{0.7467} \pm 0.0771$ |
| | POYO-[Single-session] | $\mathbf{0.9682} \pm 0.0111$ | $\mathbf{0.9194} \pm 0.0185$ | $\mathbf{0.7800} \pm 0.0702$ |
| **Pre-trained** | POYO-mp + Unit ID | $0.9675 \pm 0.0079$ | $0.9012 \pm 0.0271$ | $0.7759 \pm 0.0471$ |
| | POYO-mp + Finetune | $\mathbf{0.9708} \pm 0.0116$ | $\mathbf{0.9379} \pm 0.0193$ | $\underline{0.8105} \pm 0.0561$ |
| | POYO-1 + Unit ID | $0.9677 \pm 0.0096$ | $0.9028 \pm 0.0248$ | $0.7788 \pm 0.0548$ |
| | POYO-1 + Finetune | $\underline{0.9683} \pm 0.0118$ | $\underline{0.9364} \pm 0.0132$ | $\mathbf{0.8145} \pm 0.0496$ |

Table 2: *Behavioral decoding results across neural recordings from two nonhuman primates performing two different tasks.* All the baselines and the single-session model are trained from scratch, while POYO-mp and POYO-1 are pretrained. The standard deviation is reported over the sessions. The number of sessions in each dataset is contained in $(\cdot)$ and the top performing models in each category are indicated in boldface (first) and underlined (second).

This model achieves an average test $\mathrm{R}^2$ of 0.9512 on the center-out tasks and 0.8738 on the random target tasks, which is a marked improvement over the single-session average. When we compared the performance of our multi-session model to single-session models head-to-head (see Figure 2C-D), we found that across the board, multi-session training improves over single-sessions, and we observed even greater improvements for datasets with fewer trials (indicated by the size of the circles in the plot). On RT sessions, we observe an even bigger improvement over the single session models. These results suggest that there are significant benefits in joint-training across multiple sessions, especially when decoding more complex behaviors like the random target task.

**Scaling analysis.** By enabling multi-session training, we can start to ask questions about the extent to which having more data or more parameters can improve decoding (Figure 3). Thus, we studied the improvements obtained for three different depths of models, with L=6 (3.8M), L=14 (7.4M), and L=26 (13M) layers (parameters). For both CO and RT tasks, we find that even at the same depth (L=6), our 1 multi-session model shows an improvement over the average performance of 100 single-session models. We see further improvements with increasing the model depth, with the RT task benefiting even fur-

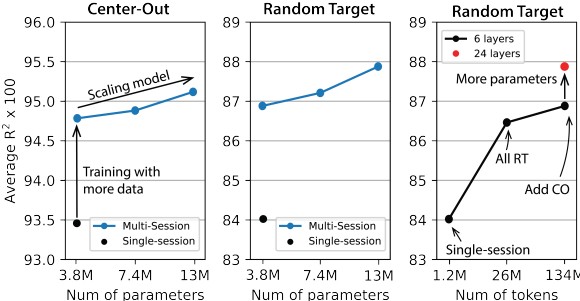

Figure 3: *Scaling curves.* $\mathrm{R}^2$ is evaluated and averaged over the test splits on both CO and RT tasks. The single-session performance is the mean over 100 different models.

ther from scaling. When we fix the model depth to L=6, and study how training jointly with more data contributes to improvement in performance, we find an improvement in performance when comparing single-session training (average 1.2M tokens) to training jointly on all RT sessions (26M tokens). When we also train jointly with other CO sessions (134M tokens in total), and increase model depth to help in accommodating the more growing and diverse training set, we continue seeing improvement in the decoding performance on RT sessions, accumulating to 4% over our single-session baseline.

### 3.4 Transferring to new sessions

After pretraining, we can then test on new sessions with unknown neurons using either the (i) *unit identification* approach we described in Section 2.5, or (ii) full *finetuning* of the model weights. Recall that when we use a unit identification approach, we freeze the weights of the model and only learn new unit and session embeddings.

**Results on held out sessions from the same animals.** We first tested the unit identification approach on held-out sessions from Monkey C that are unseen during training (Table 2, Left). In this case, we don't have correspondence between units in the training and the testing conditions. Surprisingly, we find that we can achieve comparable performance with unit identification on the pretrained model (0.9675) with that of the single-session models trained fully from scratch (0.9682). With further finetuning of the rest of the model's weights, we can improve the accuracy further (0.9708) to go beyond the accuracy of the single-session models. This highlights the robustness

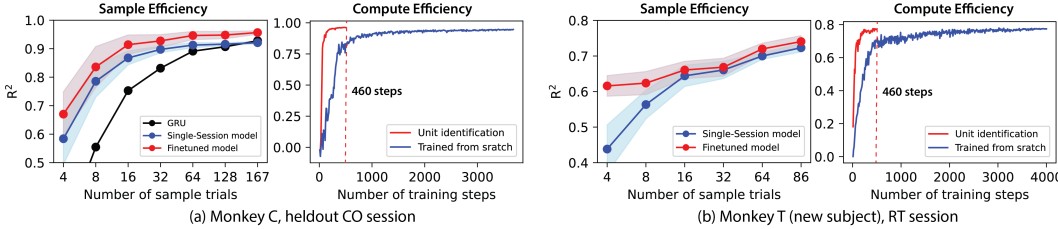

(a) Monkey C, heldout CO session      (b) Monkey T (new subject), RT session

Figure 4: *Sample and compute efficiency for training, unit identification, and fine-tuning approaches.* On the left, we show the sample and compute efficiency for a heldout CO session. On the right, we plot the sample and compute efficiency for a new animal not seen during training.

of our model and its flexibility to accommodate fresh data with even a simple input mapping (unit identification).

We compare with the single session model and a GRU baseline (Figure 4A) with our multi-session model trained on the same number of trials. Both our single-session and finetuning approach achieve good performance scaling as we increase the number of samples, with the finetuning maintaining a gap over the single session models over the entire sampling space.

**Results on a new animal performing the same tasks.** Next we tested our model on 12 sessions from a completely new animal (6 CO sessions, 6 RT sessions from Monkey T) that was not included in the training of the model (Table 2, Right). When we applied our unit identification approach on the multi-session model (POYO-mp + Unit ID), we see a bit of a dip in overall accuracy from the single-session model (POYO-[Single-session]); however, when we fine-tune the weights of the model further, we achieve an accuracy of 0.9379 on CO, which is a significant improvement over all of the other baselines. For the RT task, the single session model is 0.7569 and with unit identification we achieve 0.7669 and with full fine-tuning we get up to 0.7916 (Figure 4B). These results are promising and show that we can use our pretrained model on new animals with only a few minutes of labeled data.

**Performance on the Neural Latents Benchmark (NLB).** To understand how well our pre-trained model performs on data collected from new animals performing novel tasks with different equipment (example: touch screen vs. manipulandum), we applied our pretrained model to the MC-Maze (Monkey L) and MC-RTT (Monkey I) datasets from the NLB (Table 3)[27]. The NLB serves as a benchmark for neural representation learning and decoding and thus we can include other single-session baselines to our comparisons, including self-supervised models AutoLFADS [31] and NDT [32] which produce denoised firing rate estimates over which we fit a linear layer (+ Linear), a supervised variant of NDT

| | Method | NLB-Maze | NLB-RTT |
|---|---|---|---|
| From scratch | Wiener Filter | 0.7485 | 0.5438 |
| | GRU | 0.8887 | 0.5951 |
| | MLP | 0.8794 | **0.6953** |
| | AutoLFADS + Linear [31] | 0.9062 | 0.5931 |
| | NDT + Linear [32] | 0.8929 | 0.5895 |
| | NDT-Sup [33] | 0.8708 | 0.4621 |
| | EIT [33] | 0.8791 | 0.4691 |
| | POYO-[Single-session] | **0.9470** | 0.6850 |
| Pretrained | POYO-mp + Unit ID | 0.8962 | 0.7107 |
| | POYO-mp + Finetune | 0.9466 | 0.7318 |
| | POYO-1 + Unit ID | 0.9329 | 0.7294 |
| | POYO-1 + Finetune | **0.9482** | **0.7378** |

Table 3: *Behavioral decoding results for datasets from the Neural Latents Benchmark [27].* Best performing model is in bold and second best model is underlined.

[33] (NDT-Sup), and EIT [33]. Both datasets contain single sessions from new animals performing movement tasks that we haven't seen during training.

On the NLB-Maze dataset, we obtain a R2 of 0.8952 after unit identification, which is competitive with the baselines. These results are surprising since we do not modify the model's weights, and yet our pretrained model yields competitive results on a dataset collected under very different conditions. When we finetune the model, we boost the performance even further establishing a 4.4% gap over the best baseline. Similar trends can be observed for the RTT task (Monkey I), with even larger (2%) improvement after finetuning.

### 3.5 POYO-1: A multi-lab, multi-task model for neural decoding

Given the impressive transfer results of POYO-mp to datasets from different labs, we ask whether we can use our approach to build a model that spans even more diverse recording setups that we expect

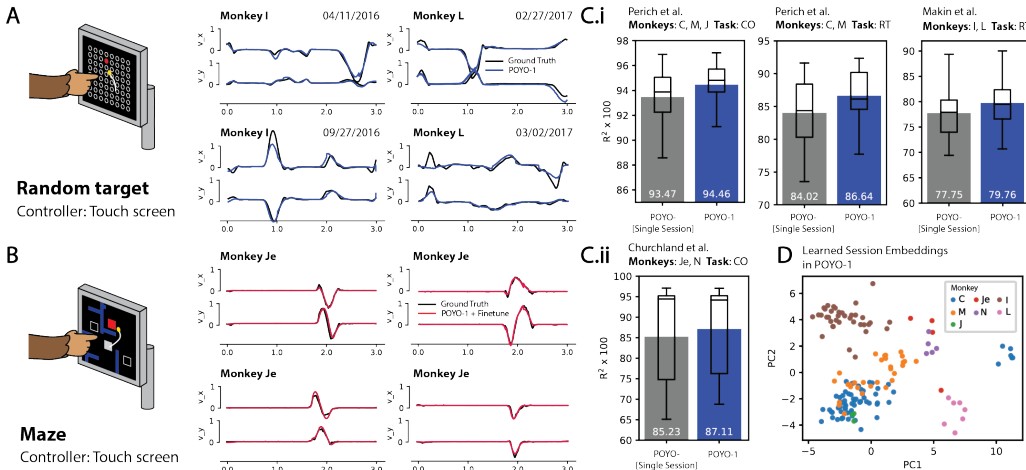

Figure 5: *Scaling up to more tasks and diverse neural and behavioral recording conditions.* (A) Random target task from Makin et al. included in training of `POYO-1` and (B) Maze task from NLB heldout for transfer testing. In (C) decoding accuracy for the single-session (gray) and the `POYO-1` model (blue). (D) PCA projection of the learned session embeddings.

to encounter when trying to unify data from many sources. In total, we used datasets from seven non-human primates spanning three different labs, with a total of 27,373 units and 16,473 neuron-hours for training our model. We call this pretrained multi-lab, multi-task model `POYO-1`.

Even in light of the high amounts of variability across these different datasets, `POYO-1` provides consistent improvements over the single-session models (Figure 5C). When tested on a number of different transfer tasks (Tables 2, 3), we again find that the unit identification and finetuning approaches provide effective strategies for adapting our pretrained model to new datasets. We obtain notable performance on the NLB-Maze, where we find that we obtain a $R^2$ of 0.9329 with only unit identification remapping, an almost 4% improvement over the unit identification result for our `POYO-mp` pretrained model, suggesting that we cover more of the task space with `POYO-1`.

When comparing the `POYO-1` model with `POYO-mp` model, it is clear that both methods have their strengths. `POYO-mp` excels on the datasets sourced from the same lab, with 0.8788 on RT sessions compared to `POYO-1`'s 0.8664. On the other hand, both models exhibit great transfer capabilities, with `POYO-1` having the edge especially when using unit identification, indicating its ability to generalize better across diverse experimental conditions. This flexibility and scalability make these methods promising tools for future research in analyzing neural data from diverse sources.

**Visualizing the session embedding space.** We visualized the learned task embeddings to see if the model learns relationships between sessions seen during training (Figure 5D), even though it was not explicitly trained with this information. This analysis revealed clusters corresponding to different data sources and tasks, suggesting that our model has not only learned to identify patterns within each session but also recognized overarching structures across sessions. In particular, we find that the datasets collected in the Miller Lab (C,M,J) used to train the `POYO-mp` model are mapped into similar regions of the session latent space, and (I and L) sessions are mapped to their own distinct clusters.

## 4  Related Work

**Transformer architectures for time-series data.** Transformers have emerged as a powerful model for processing time-series data due to their ability to capture long-range dependencies [34, 35]. The key to their success lies in the self-attention mechanism [14], which allows the model to weigh the importance of each time step in the input sequence when producing an output. A typical approach in applying transformers to time-series data involves discretizing the continuous time domain into discrete bins [35], and treating the binned data as a sequence of tokens. Each bin is then linearly projected into a high-dimensional embedding, which is input to the transformer model. The Perceiver framework [18, 13] moves away from the idea of patches, by directly processing the input bytes,

using cross-attention layers to reduce the complexity of attention. In addition to regular timeseries, there has been interest in applying transformers to model irregular timeseries including event stream and point process data [36, 37, 38]. However, many of these models work on univariate event streams and when they extend to multivariate cases, they assume fixed and known input channels.

**Transformers applied to neural data.** With the recent advances in transformers for sequence modeling in many different time-series, transformers have also recently found successful applications in building representations of neural population activity [32, 33, 39]. In these models, the spiking activity is first binned and then the estimated bin counts are tokenized. In the neural data transformer (NDT) model [32], the firing rates of all neurons in the population are embedded jointly in one token (time step or bin). In the embedded interaction transformer (EIT) [40], neurons are considered independently in one stage of processing and at a population level in a second stage, and thus the whole dataset is tokenized over both neurons and time. In the spatiotemporal (STNDT) model [39], two different tokenizations are also considered, one in space and one in time, and two representations are learned jointly for both tokenizations. In all cases, binned data are used and the models are trained on a single session and fixed set of neurons.

**Multi-session training and alignment.** The idea of decoding across multiple sessions has been explored in previous work [41, 42, 43, 44]. In many of these works, an initial baseline representation is formed on one day and alignment-based approaches are used to transfer a model trained on one session across recording days [23, 45, 12, 46, 47]. A subset of these methods [41, 43] can be trained on many sessions jointly, but rely on the assumption of shared dynamics or structure of a single task to achieve alignment. To the best of our knowledge, our work is the first to demonstrate multi-session transfer across subjects performing different tasks, and the first to demonstrate scaling across different data and model sizes.

# 5 Discussion

In this paper, we introduce a novel framework for training transformers on large multi-session, multi-task neural activity datasets. To tackle the challenges of training on such large heterogeneous sources, we introduce a novel spike-level tokenization scheme and architecture that enables the model to learn from populations with varying numbers of neurons. We show that training a single unified model on multiple recordings is possible, and find that it leads to improved decoding performance. Finally, we build two large pretrained models (POYO-1, POYO-mp) that can be efficiently fine-tuned on new datasets, and make them available as a resource to the community.

In contrast to models trained on a single dataset, the pretrained models that we have developed provide a potential way to compare and contrast datasets, and also understand common motifs of activity and dynamics that may be shared across different sessions, tasks, and individuals. Thus, it will be critical to develop tools to probe the patterns and motifs learned by such models and characterize the neural mechanisms underlying different tasks and computations. In particular, we look to understand how spike tokens are grouped across different latent tokens and how the dynamics of the population are modeled in this latent space. Additionally, our proposed unit embedding space allows us to map units into a high-dimensional space; thus understanding how unit projections are organized might help reveal the similarities between different neurons and the nature of their interactions. Similarly, we can analyse the session embeddings to glean insights into inter-session and across-animal differences.

Our work shows how pretraining on diverse data, including datasets from animals performing different tasks and across different laboratories, can all help to improve our ability to decode from novel and unseen neural datasets. Already, our results demonstrate the positive effect of scale for neural data analysis. However, to scale this approach further and integrate even more diverse brain regions and tasks, it will be critical to move toward a self-supervised objective. Thus, our current architecture and multi-session framework could be also extended to self-supervised tasks like generative next-event prediction or masked modeling to allow for even larger datasets to be ingested.

This framework has the potential to advance neuroscience research in several ways. By enabling the development of large pretrained models that can be fine-tuned for various downstream tasks, our work can accelerate progress in brain-machine interfaces and disease modeling applications. The ability to quickly and effectively decode neural activity from diverse datasets and modalities can have significant implications for improving our understanding of the brain and developing new therapeutic interventions.

## Acknowledgements

We would like to thank Patrick Mineault for providing a lot of great feedback on the work, and Jingyun Xiao for helping to run baselines for NLB. This project was supported by NIH award 1R01EB029852-01, NSF award IIS-2146072 as well as generous gifts from the Alfred Sloan Foundation (ED), the McKnight Foundation (ED), and the CIFAR Azrieli Global Scholars Program (ED). BAR acknowledges support from NSERC (Discovery Grant: RGPIN-2020-05105; Discovery Accelerator Supplement: RGPAS-2020-00031; Arthur B. McDonald Fellowship: 566355-2022) and CIFAR (Canada AI Chair; Learning in Machine and Brains Fellowship). MGP acknowledges support the Fonds de recherche du Quebec Santé (Grant: Chercheurs-boursiers en intelligence artificielle). GL acknowledges support from the Canada CIFAR AI Chair program, the Canada Research Chair for Neural Computation and Interfacing, and NSERC Discovery grant RGPIN-2018-04821.

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

# Appendix

## A  Additional Model Details

### A.1  Rotary position encoding

Rotary Position Embeddings (RoPE) are used to incorporate relative positional information into transformer models in a memory and compute efficient manner[19]. Unlike absolute position embeddings which are added or appended to the input embeddings, these are directly incorporated in the attention mechanism.

First, we define a $2 \times 2$ rotation matrix $\mathbf{R}_{2\times2}(t, T)$ for a given timestamp $t$ and time-period $T$:

$$\mathbf{R}_{2\times2}(t, T) = \begin{bmatrix} \cos(2\pi t/T) & -\sin(2\pi t/T) \\ \sin(2\pi t/T) & \cos(2\pi t/T) \end{bmatrix}$$

$\mathbf{R}_{2\times2}(t, T)$ provides a time/positional encoding with a time period $T$. Intuitively, the sinusoidal values in this rotation matrix can help the model learn and distinguish variations in timestamps at a timescale $T$.

Given the head dimension of the attention block $D$, we create a $D \times D$ rotary embedding matrix $\mathbf{R}(t)$ for a timestamp $t$ by constructing a block-diagonal matrix of rotation matrices $\mathbf{R}_{2\times2}$, where the time-periods are varied logarithmically between $T_{min}$ and $T_{max}$

$$\mathbf{R}(t) = \begin{bmatrix} \mathbf{R}_{2\times2}(t, T_0) & 0 & \cdots & 0 \\ 0 & \mathbf{R}_{2\times2}(t, T_1) & \cdots & 0 \\ \vdots & \vdots & \ddots & \vdots \\ 0 & 0 & \cdots & \mathbf{R}_{2\times2}(t, T_{\frac{D}{2}-1}) \end{bmatrix}$$

Given $D$, $T_{min}$, and $T_{max}$, we generate $T_i$ by the following equation:

$$T_i = T_{min} \left( \frac{T_{max}}{T_{min}} \right)^{2i/D}$$

In our models, $T_{min}$ and $T_{max}$ were set to $1ms$ and $4s$ respectively.

This rotation matrix is incorporated into the attention mechanism by replacing the attention score calculation with the following equation:

$$\mathbf{a}_{ij} = \text{softmax}\big( \left(\mathbf{R}(t_i)\mathbf{q}_i\right)^T \left(\mathbf{R}(t_j)\mathbf{k}_j\right) \big)$$

Since $\mathbf{R}(t_i)^T\mathbf{R}(t_j) = \mathbf{R}(t_j - t_i)$, the attention score $\mathbf{a}_{ij}$ depends only on the relative timing between the query and key tokens. This is what makes RoPE relative in nature.

What is explained above has been introduced in [19], and use successfully in many models [18]. In our experiments, we found an interesting extension of RoPE. In addition to incorporating RoPE in calculating the attention scores, we also incorporate it in the weighted-summation of values inside the attention mechanism. We do this to encode the relative timing between values and queries in the output of the attention block. A direct, but computationally inefficient, way of doing so would be:

$$\mathbf{z}_j \leftarrow \mathbf{z}_j + \sum_i \mathbf{a}_{ij}\mathbf{R}(t_i - t_j)\mathbf{v}_i$$

This is inefficient since it requires us to explicitly compute $\mathbf{R}(t_i - t_j)$ for all $N^2$ values of $(t_i - t_j)$. We use the fact that $\mathbf{R}(t_i - t_j)$ can be factorized as $\mathbf{R}(-t_j)\mathbf{R}(t_i)$ to reach the following, much more efficient, implementation:

$$\mathbf{z}_j \leftarrow \mathbf{z}_j + \mathbf{R}(-t_j) \sum_i \mathbf{a}_{ij}\mathbf{R}(t_i)\mathbf{v}_i$$

That is, we first apply rotation matrices $\mathbf{R}(t_i)$ to all values $\mathbf{v}_i$ before passing them to as inputs to the attention block. Then we apply the rotation matrices $\mathbf{R}(-t_j)$ to its outputs.

This *value-rotation* mechanism captures relative-timing information more explicitly in the output of the attention block, allowing the feedforward network to take advantage of timing information. In the absence of this mechanism, the only way timing information is added to the latent tokens is through the attention scores. We observed in early experiments that our model performed better with value-rotation enabled for the input cross-attention and all self-attention layers.

As discovered in the Perceiver framework [13], we observed an improvement in performance upon rotating only half of the head dimensions. That is, half of the $\mathbf{R}_{2 \times 2}$ matrices in $\mathbf{R}$ were replaced by $2 \times 2$ identity matrices.

## A.2 Time-based context window

We consider an arbitrary window of time $[t, t + T]$, meaning that we do not use the trial structure when training our model. In our experiments, we set $T$ equal to 1 second. At inference time, we use a sliding window of step $500$ms and average the predicted output in overlapping segments.

In language models, and more generally transformer-based models, the context window is defined with respect to the number of tokens. The neural data modality is a bit more complex, as the number of spikes scales with the number of units: Across multiple datasets, we record a variable number of units, ranging from tens of units to hundreds of units. Defining the context window with respect to number of tokens will result in a variable time duration, long for small populations and short for larger ones. Our context window is defined with respect to time (fixed to $1s$), each sequence will have a variable number of tokens ($M$). When creating batches, we pad to the largest sequence in the batch, and appropriately use an attention mask in the first cross-attention layer. GPU hardware is optimized to process fixed-size tensors, to make computation more efficient, we introduce a distributed training load balancing mechanism, which deals with variable-length input sequences (Appendix A.4.4).

## A.3 Further details on tokenization

**Delimiters.** When extracting the spike data in a window of time $[t, t + T]$, we include delimiters for each unit, indicating to the model that we are tuning into the activity of that unit between times $t$ and $t + T$. This choice is important because: 1) we are using relative position encoding, so the beginning or the end of the window is not clear, and 2) our tokenization only captures activity and not inactivity. If we consider a unit that is inhibited and does not fire in that window of time, the model would be unaware of the presence of that unit, unless we add [START] and [END] tokens. We learn two embedding vectors for these delimiters $\mathbf{x}_{start}$ and $\mathbf{x}_{end}$. For each unit in the population, we add these two tokens with embedding equal to the sum of the delimiter embedding and the unit embedding, and assign each token timestamps $t$ and $t + T$ respectively.

**Latent tokens.** Recall that we divide the latent tokens into groups of $n$ and spread each group uniformly over the context window. Specifically, we use a total of 256 latent tokens and divide them into groups of 8, this means that there will be 32 tokens that share the same timestamp. We experiment with weight-sharing, where we set the learned latent tokens in the same group to have the same weights ($\mathbf{e}_0, \ldots, \mathbf{e}_{31}$ in Fig 1). This means that tokens in the same group (size 8) will have the same embedding $\mathbf{z}_{0,i} = \mathbf{e}_{(i \bmod 32)}$, but different timestamps ($t_0^l, \ldots, t_7^l$ in Fig 1). We believe this is a reasonable inductive-bias to architecturally enforce since, intuitively, we want all sets of latent tokens that share a timestamp to query the spike tokens in the exact same manner, just with a different timestamps. These tokens will evolve independently once the information is pulled from the input space (after the first cross-attention).

## A.4 Training details

The model is trained using the LAMB optimizer [28] with weight decay. The learning rate is held constant, then decayed towards the end of training (last 25% of epochs), using a cosine decay schedule. Single-session models are trained on a single GPU with a batch size of 128 while large models are trained with 8 GPUs with a total batch size of 1400. Note that we didn't see any benefits in increasing the batch size when training single-session models.

### A.4.1 Compute

The large models are trained on a machine with an AMD EPYC 7452 32-Core Processor and 8 Nvidia A40 GPUs (48Gb memory), `POYO-mp` was trained for 2 days, and `POYO-1` was trained for 5 days (both for a total of 400 epochs). Single-session models are trained with a single Nvidia GeForce RTX 3090 GPU, and take less than an hour to train. Finetuning models is also done with a single GPU. Unit identification converges very quickly and takes less than a minute on a single GPU or a few minutes on the CPU.

### A.4.2 Data augmentation

Previous work [48, 49] explores the use of augmentations when training neural population models. In our experiments, we use the unit dropout augmentation which randomly samples a subset of the population. We set a minimum population size of 30 for this augmentation to ensure that it is not too destructive (note that our model is still trained with any number of units, this limit is only imposed for the augmentation). We did not explore the use of other augmentations, but believe it could be a promising direction to further improve the capabilities of the model.

### A.4.3 Loss

We train our model using a mean-squared error loss over the hand velocity sequence. We normalize the velocity data using z-scoring at the lab level, this means that all velocity data in datasets from the same lab is rescaled using the same scalar.

**Dealing with variable sampling rate.** When training on mixtures of datasets from multiple labs, any given batch will contain behavior sequences that have different sampling rates. In a window of 1s, some sequence will have more timepoints over which we will evaluate the MSE compared to other sequences. To avoid the over-representation of samples that simply have a high sampling rate, we evaluate the error on a randomly sampled 100 timepoints (these points are randomly sampled at each pass).

**Weight in training mix.** While behavior during random target tasks can be more complex and noisy, behavior during center-out reaching tasks is more structured, the monkey usually gets a preparation phase, where the movement can be planned [20]. The latter task has been studied extensively for that reason. Knowing that neural activity is very salient during center-out reaching, we increase the weight of the prediction loss during the reaching segments by a factor of 5. The idea of over-weighing samples of "good quality" is not novel and is commonly used when training language models [15].

### A.4.4 Load Balancing

We distribute our training over multiple GPUs. Note that, in our experiments, the input sequence length ($M$) can vary anywhere between 1k and 20k tokens. Because of the variability in the length of the input sequences, we distribute sequences that have close length to same GPU node.

We now provide further details on our load balancing strategy:

- Each GPU node has a local batch size. A GPU that is assigned large sequences will have a smaller local batch size than one that is assigned shorter sequences.
- We create a number of data buckets equal to the number of GPUs. Each bucket $i$ is assigned sequences that are of length $M \in [M^i_{min}, M^i_{max}]$, and a local batch size $B_i$.
- We select these parameters in such a way that we effectively utilize all available compute capacity, and minimize the amount of padding used.

### A.5 Evaluation Details

During training, we train on any 1s segment of the recording, but at evaluation we report the decoding performance using the same evaluation strategy used in previous work [27, 10]: for center-out reaching datasets, we report the decoding score during the reaching movement (only when the trial is completed successfully), for random target datasets that are trialized (hold period followed by 3/4 random reaches), we report the decoding score during the reaching as well, and finally for all other continuous random target datasets, we report the decoding score for all segments. Note that none of the segments used for testing are seen during training.

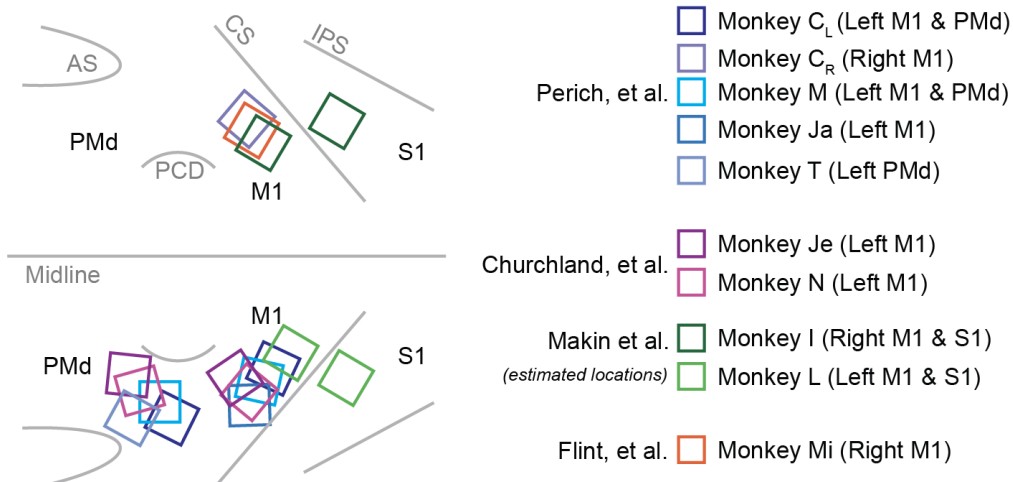

Figure 6: Diversity of neural recordings used to train POYO. We visualized the array locations for all of the datasets used in our current study. POYO-mp is trained on recordings from PMD and M1 (CL, CR, M, Ja) and POYO-1 is trained on recordings from (CL, CR, M, Ja, I, L, Mi) spanning PMD, M1, and S1 across both hemispheres and in seven animals.

## A.6  Fine-tuning Details

When finetuning the model, we perform gradual unfreezing of the weights. For the first few steps, we optimize the unit embeddings and the session embedding only (unit identication) and then unfreeze the rest of the weights. We use the LAMB optimizer with batch size 128, learning rate $10^{-4}$ and weight decay $10^{-4}$. We use the exact same hyper parameters and finetuning process for all experiments we report. This strongly suggests that our method is general, and easy to adapt to new data, without the need for expert neural network training knowledge, or expensive hyperparameter search.

## B  Datasets

To train and evaluate our framework, we curated a large number of open and publicly available datasets that represent a range of neural and behavioral recording platforms.

### B.1  Datasets used for training

**Datasets used to train** POYO-mp**.** To build our first large-scale model (POYO-mp), we acquired 100 sessions of unique recordings from three non-human primates performing two different movement tasks that were previously studied in four publications [21, 22, 20, 23]. We make these datasets publicly available on the DANDI open repository [29]. In all of these datasets, a nonhuman primate is seated in a primate chair and executes movements using a custom 2-D planar manipulandum to control a cursor on a computer screen. They performed one of two tasks within a session.

- *Center-out Task* (CO): They performed a standard center-out reaching task involving eight outer targets arranged around a circle with a radius of 8 cm. The monkeys were required to move to the center target, wait for a variable hold period, and then move to one of the outer targets upon receiving an auditory go cue. The task aimed to study neural activity during movement planning, preparation, and execution.

- *Random Target Task* (RT): The setup is similar, the targets are not arranged around a circle but are randomly placed. The monkeys were required to move between 4 targets upon receiving an auditory go cue.

After extensive training, the monkeys were implanted with chronic multi-electrode arrays in the primary motor cortex (M1) and the dorsal premotor cortex (PMd). Neural activity was recorded using

a Blackrock Cerebus system, and the data was manually processed offline to identify single neurons. To ensure only independent channels were included, steps were taken to identify and disable potential candidates with high crosstalk and to exclude cells with a high percentage of coincident spikes.

**Datasets used to train the multi-lab, multi-session model** (POYO-1)**.** When training POYO-1, we acquire more data as shown in Table 1, adding all of the datasets outlined from Churchland et al., [24], Makin et al., [25], and Flint et al., [26]. We still hold out the 12 sessions from Monkey T, 2 sessions from Monkey C, and the two Neural Latents Benchmark (NLB) [27] datasets from training.

As depicted in Figure 5, instead of using a manipulandum like in the previous experiments, these new datasets use a touch screen and have different sampling rates for their behavioral outputs. In addition, in the Churchland, et al. datasets, they use threshold crossing based processing rather than applying spike sorting to identify single units. We refer the reader to the listed publications for more details about how these datasets were collected.

We would like to reiterate that there are major differences in the way the datasets were collected. Yet, in spite of this variability, it is important that our method works without having to standardize the datasets across sites further and apply specialized techniques to one dataset but not other. Thus we did not attempt to homogenize or standardize the data. The only processing that we perform is on the behavior to clip outliers in the behavior based upon times of high acceleration. We use a simple threshold heuristic to avoid training on periods with extremely high acceleration.

This means that:

1. We do not filter units based on a presence ratio, or reject multi-units.

2. We do not re-apply the same spike sorting algorithm on all datasets, and use the data as is. In [24], the spiking events obtained through threshold crossings (i.e. all units are multi-units), and in [25], we have both threshold crossing units and spike sorted isolated single units, and we use both. Each algorithm used in each lab was tuned differently, but we consider this a good example of diversity that a general model needs to deal with.

3. We do not resample or process the velocity timeseries.

Minimizing the amount of processing needed to integrate new datasets into our model is key for more easily scaling to more datasets, and also providing an accessible model that the community at large can use out of the box, without having to adapt to a specific standard.

### B.2 Datasets for evaluation and fine-tuning

In addition to holding out 20% of the data from each session, we hold out all sessions from Monkey T (6 CO, 6 RT). We also use two datasets from the Neural Latents Benchmark [27], MC-Maze and RTT. The Maze dataset consists of recordings from primary motor and dorsal premotor cortices of a monkey performing reaches with instructed delays to visually presented targets while avoiding the boundaries of a virtual maze. This dataset offers a variety of behavioral configurations, with each configuration using a different combination of target position, number of virtual barriers, and barrier positions, resulting in a variety of straight and curved reach trajectories. With thousands of trials, the Maze dataset allows for a rich investigation into the structure of population activity, while its instructed delay paradigm enables a clear separation of neural processes related to preparation and execution. On the other hand, the RTT dataset introduces different modeling challenges, as it contains continuous, point-to-point reaches that start and end in various locations, have highly variable lengths, and few repetitions. We report the performance of our models on the same segments of data used to evaluate models in the NLB benchmark.

## C  Additional Results

### C.1  Comparison with single-session baselines

In our comparisons in Table 2, we compared POYO against existing baseline models that are commonly used for neural decoding [30]. In particular, we applied a Wiener Filter (WF), Feed Forward Network (MLP), and a Gated Recurrent Unit (GRU). These models predict the behavior at each timestamp $t$ given a history window of neural activity $[t - T, t]$. For the NLB datasets, we also compare with

Auto-LFADS [31], an unsupervised approach that can be used to obtain single trial smoothed rate estimates for a population of neurons before applying a linear layer on top of the inferred rates to obtain an estimate of the behavior. While this model isn't designed for decoding per se, it is frequently used for denoising for BMI decoding applications.

For the NLB, we use a 5 ms binning of the neural activity for training models and for evaluation of the behavioral output as this is the finest resolution in the benchmark. For the rest of the MP datasets, we use a 10 ms binning rate for the neural data and behavior as this corresponds to the behavior sampling rate for these datasets.

The hyperparameters we used for training our single-session POYO model can be found in Table 4.

| Hyperparameter | Value |
|---|---|
| Embedding Dimension | 128 |
| Head Dimension | 64 |
| Number of Latents | 128 |
| Depth | 6 |
| Number of Heads | 8 |
| FFN Dropout | 0.3 |
| Linear Dropout | 0.3 |
| Attention Dropout | 0.3 |
| Weight Decay | 1e-4 |
| Learning Rate | 4e-3 |
| Batch Size | 128 |

Table 4: Hyperparameters used for training all POYO single-session models

## C.2 Robustness of Unit-identification

Unit-identification, introduced in Section 2.5, can be seen as an approach for *identifying* units from a new dataset. To evaluate the robustness of this process, we ran unit-identification on a session that was already seen during pre-training. We test whether finetuning on this "new" set of units, will map them close to their true embedding (found during pre-training). We use our largest model, POYO-1 , start with randomly initialized unit embeddings and run unit-identification.

We perform this experiment on two different animals. In Figure 7, we show the evolution of unit-embeddings during training. To compare the newly calibrated set of units with its pre-trained version, we normalize the unit-embeddings and report the cosine similarity averaged over the set of units. Note that normalization is done because POYO's first cross-attention layer applies layer norm to these embeddings. We also report the "unit identification accuracy" which we define as the ratio of tuned unit embeddings that have their true unit embedding as their nearest neighbor. Results for these experiments are present in Table 5.

| Dataset | Num of Units | Unit-Id Accuracy | Unit-Id Cosine-Similarity |
|---|---|---|---|
| Monkey C, CO, 2013/10/03 | 73 | 1.000 | 0.845 |
| Monkey M, CO, 2014/02/03 | 116 | 0.966 | 0.802 |

Table 5: *Unit re-identification results*. Cosine-similarity measure is averaged over all units in a dataset. Accuracy is measured for 1-nearest-neighbor classifier.

Overall, this analysis suggests that the unit-identification approach is robust and reliable for identifying units. We believe that understanding the functional similarities of units that are mapped close to each other in the unit embedding space will be an interesting avenue for future work.

## C.3 Hyperparameter sensitivity

When initially training the single-session models, we selected a subset of datasets (out of the 100 sessions) and performed a random hyperparameter search and selected the set of hyperparameters with the best performance on the validation set. These experiments quickly revealed that the model was very stable to a wide range of hyperparameters. In Figure 8, we report some examples of this for

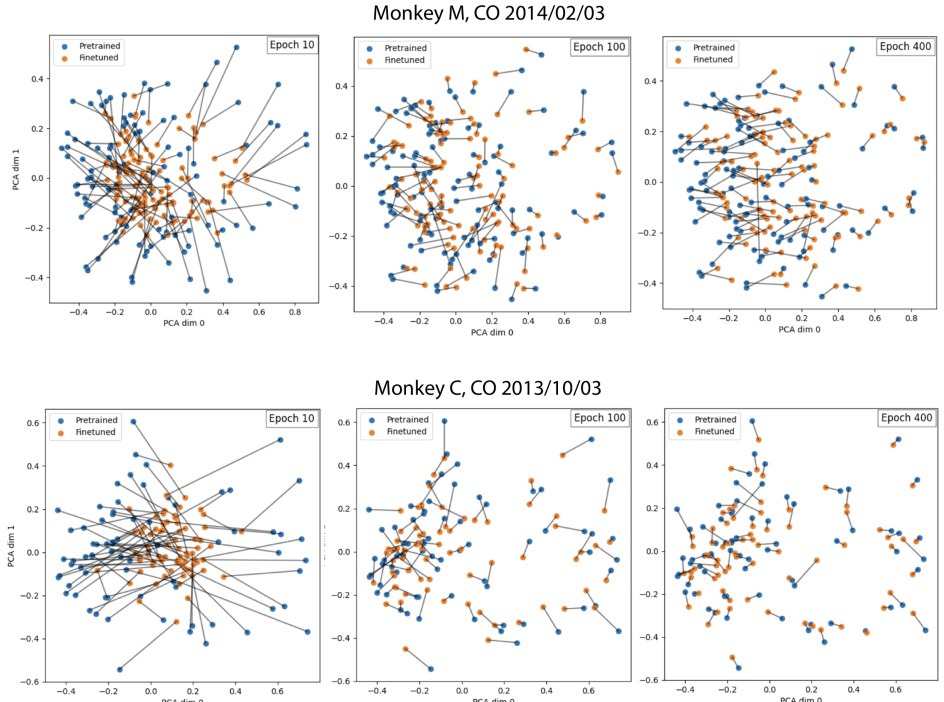

Figure 7: Evolution of the unit embeddings with unit-identification on sessions from two different animals. From left to right, the finetuned embeddings (orange) are visualized at 10, 100, and 400 epochs, relative to their true embedding in the pre-trained model (blue).

two sessions. We perform 300 runs for each session, then report the average performance for each hyperparameter.

Given these findings, we selected the best set of hyperparameters for the subset and used the same parameters when training all of the 100 single-session models that we study in Figure 2.

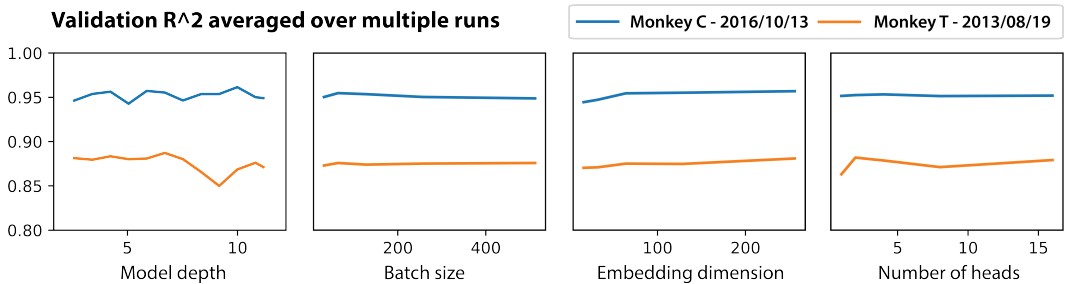

Figure 8: Single-session model performance for a wide range of values for different hyperparameters. The $R^2$ score is reported on the validation sets for both datasets.

