# OpenReview forum: "A Unified, Scalable Framework for Neural Population Decoding"
_NeurIPS.cc/2023/Conference — NeurIPS 2023 poster_

### Official Review · Reviewer_Mfux · 2023-07-03

**Soundness:** 3 good
**Presentation:** 3 good
**Contribution:** 3 good
**Rating:** 7
**Confidence:** 3

**Summary:**

Neural population decoding refers to inferring the behavioral output of organisms from a recording of a population of neuronal recordings. This paper introduces a transformer architecture to improve the accuracy of decoding. The efficiency of the system is increased by representing input spike trains with event-based tokens and by switching to a latent representation (to decrease the size of attention matrices).

In the presence of behavioral data, the model can identify new neurons (units) that are potentially obtained from different experiments. This is done by freezing the parameters of the network during gradient descent except for the encoding of the new units. This is a novel and interesting approach although it depends on the presence of behavioral recordings.

The method is tested on multiple experiments where neural activity recordings are obtained from motor cortical regions of monkeys while the monkeys are engaged in tasks involving arm movements.

**Strengths:**

- A novel framework for neural population decoding
- Efficient representation of the recordings via event-based tokenization
- Identification of units from new experiments in a way that is consistent with previous experiments. i.e., embedding a new unit such that the embedding location is consistent with the locations of other units in previous experiments whose activities/roles are similar to the new unit.
- Use of multiple experiments done in different labs to demonstrate the method.
- The inference accuracy of the method seems to improve significantly upon strong baselines.

**Weaknesses:**

- I oppose alluding to 'foundation models', which have become popular in natural language processing. Importantly, such models are trained in an unsupervised manner and demonstrate emergent abilities to solve multiple downstream tasks. The proposed architecture depends strongly on the presence of joint behavioral recordings. Similarly, it wouldn't make sense to apply this trained model to recordings from, say, the visual cortex.

- [line 165] Are commands given via natural language? If not, please provide mathematical description instead (or in addition).

- One simple experiment to probe the fidelity of "unit identification" could be: (i) take a recording, learn embeddings of units, etc. (ii) shuffle the order of units and repeat the experiment by adding rows to Embed as described in text. (iii) Compare and report the similarity between the two embeddings.


**Questions:**

- What is the relationship between the context window T (1s) and Tmax (4s) used in positional encoding? How does Tmax change with respect to T?

- Could you provide more details on extending the model to self-supervised tasks (mentioned at the end of the paper)?

**Limitations:**

Yes.

---

> ### Author Rebuttal · Authors · 2023-08-10
>
> Thank you for your feedback and questions! We are very excited to hear that you found the paper to be “novel” and also appreciate our use of “multiple experiments done in different labs”.
>
> In what follows, we will provide point-by-point replies to your questions.
>
> > 1. “I oppose alluding to 'foundation models', which have become popular in natural language processing. Importantly, such models are trained in an unsupervised manner and demonstrate emergent abilities to solve multiple downstream tasks. The proposed architecture depends strongly on the presence of joint behavioral recordings. Similarly, it wouldn't make sense to apply this trained model to recordings from, say, the visual cortex.”
>
> **Reply:**  We agree with the reviewer that our current model wouldn’t be considered a foundation model, and we will clarify that in the text. However, POYO makes important advances towards a foundation model for neuroscience on multiple fronts.
>
> 1. Scalability:  Building foundation models for neural data requires training on large amounts of data, and a key innovation of our work is developing a framework where integration and training on many sessions is made possible.
>
> 2. Pre-training and transfer learning:  A key advantage of large language and other foundation models is that they enable fine tuning on smaller datasets. In our experiments, we show how our pretrained models POYO-MP and POYO-1 can be used on a number of diverse recordings from non-human primates, even if the data is from a new animal, their behavioral task is different (Figure 2), and when the preprocessing of the data is different (POYO-MP to NLB-Maze). In our few-shot learning experiments, we show that finetuning can quickly integrate new sessions from different labs with fewer than 32 labeled trials comprising less than 2 minutes of recordings.
>
> 3. Democratizing AI:   In addition to providing pretrained models and transfer strategies, POYO also can be very easily adapted and trained without extensive hyperparameter tuning (which usually requires expertise), and without need for large compute resources. Unit identification can be done on CPU in a few minutes, or finetuning can be done on a single GPU. We note that the same hyper parameters are used across all our results in table 2, showing its robustness.
>
> Moreover, as noted in the paper, there is no reason that the POYO tokenization scheme and architecture could not be used for self-supervised learning. We will mention this in the text when we raise the issue of foundation models.
>
> > 2. “Are commands given via natural language? If not, please provide a mathematical description instead (or in addition).”
>
> **Reply:** The command is not given via natural language, we will add this for clarity. We will update the text to include a clear mathematical formulation, to avoid any confusion:
> A query token is defined by its embedding $y_{0, i} = Embed(session\ id) + Embed(behavior\ id) $ and its timestamp $t_i$.
>
> > 3. One simple experiment to probe the fidelity of "unit identification" could be: (i) take a recording, learn embeddings of units, etc. (ii) shuffle the order of units and repeat the experiment by adding rows to Embed as described in text. (iii) Compare and report the similarity between the two embeddings.
>
> **Reply:**  Thank you for suggesting this experiment! We provide an in depth discussion of the results in the General Response (see Table 2). Further visualizations are provided in the attached PDF (See Figure 1).
>
> As you can see, when we tested your idea using POYO-1 weights, we indeed found that the tuned embeddings converge to their original unit embeddings (in a nearest-neighbor sense). We are excited to refine this analysis and include these results in the final paper. Thanks again for your suggestion!
>
>
> > 4.  What is the relationship between the context window T (1s) and Tmax (4s) used in positional encoding? How does Tmax change with respect to T?
>
> **Reply:** This is a great question that we explored in our initial experiments, where we did a sweep over various values of both Tmin and Tmax for POYO-SS (single session) and found that various values of Tmax provided good performance. We hypothesize that as long as Tmax >= T, our model should perform well. We will update appendix C.3 to include these insights.
>
> > 5. “Could you provide more details on extending the model to self-supervised tasks (mentioned at the end of the paper)?”
>
> **Reply:** The PerceiverIO architecture that we build upon is flexible and allows for different predictive targets, which could be adapted to other objectives like the prediction of the firing activity of held out and masked neurons as in the NDT.  Contrastive learning can also be used to generate various views (different sub-populations for example) and then learn an embedding that maps these views to similar parts of the latent space (i.e., using BYOL or SimCLR). We plan to add more discussion on these potential directions for future work in the revised manuscript.

---

> > ### Comment · Reviewer_Mfux · 2023-08-13
> > **Thank you for the experiment on unit identification.**
> >
> > These results will help quantify the extent of unit identification and improve the paper. Thank you.
> >
> > Thanks also for the other clarifications.
> >
> > The term "foundation model" is overloaded and means a different thing in NLP. While I agree with the authors' scalability, pre-training, etc. claims, I think considering this as a foundation model (or something along the way to a foundation model) will only further confuse the field. I remain strongly opposed to alluding to it in the main text, except for the Discussion, where those claims could be mentioned and it would be appropriate to suggest that foundation models also have such properties (and more).
> >
> > I think this is a good paper, and I decided to keep my score at 7.

---

> > > ### Author Response · Authors · 2023-08-19
> > >
> > > Thank you for your positive response, and your thought-provoking questions. We definitely agree that the term “foundation models” needs to be discussed in more detail. To avoid any confusion, we have removed the use of “foundation models” in the Results. In the discussion, we will clarify what aspects of our model are novel and of interest to the neuroscience community, and what remains to be done in future work.

---

### Official Review · Reviewer_BY6C · 2023-07-04

**Soundness:** 4 excellent
**Presentation:** 3 good
**Contribution:** 4 excellent
**Rating:** 7
**Confidence:** 4

**Summary:**

Deep learning and transformer models have shown great promise in identifying structure from large datasets. With recent advancements in neural recording methods, it is now possible to generate rich and heterogeneous recordings from large populations of neurons across multiple brain regions and experimental conditions. The proposed work bridges this gap by introducing a new approach to modeling neural dynamics across these recordings. The authors leverage transformer models by tokenizing neural activity, preserving the temporal structure by treating each spike as a discrete token. This strategy allows training across multiple sessions and individuals. The success of the approach was extensively tested in several datasets across multiple labs, brain regions, and tasks, demonstrating increased decoding performance compared to alternative methods.

**Strengths:**

The paper is presented clearly and is technically sound. The method was tested and shown to work well across multiple datasets. The proposed strategy leverages the success of transformer models and applies it to neuroscience by introducing a novel way to tokenize neural activity. Extracting shared variability across multiple datasets is crucial for neuroscience and biomedical applications, and this approach could provide a framework for analyzing these emerging datasets. The authors demonstrated the success of the approach when trained on multiple neural recordings and tasks, with the ability to pool data across sessions and animals. Decoding results showed that the method outperforms alternative approaches, including lesion versions of the introduced model, when tested within and across sessions and animals. They also investigated the impact of different architectures and numbers of parameters on performance. Importantly, they showed that minimal fine-tuning allows the model to improve decoding performance on other tasks. The proposed strategy could lay the foundation for studying unifying principles of neural computation.

**Weaknesses:**

However, it should be noted that this strategy heavily relies on having access to multiple large neural recordings, which may not be feasible for most basic neuroscience research where datasets are often limited. While the authors tested the method across multiple brain regions and tasks, all of them were motor tasks in motor-related areas. While this choice likely facilitated the identification of shared structure and enabled transfer learning, it would be even more intriguing if this approach could be applied across categorically different tasks, such as sensory encoding and decision-making, spanning from sensory areas to higher cortical regions. This could potentially help identify universal coding principles of neural computation as well as task-specific coding principles related to motor function.

**Questions:**

I would be interested in applying this method to even more diverse recordings throughout the brain and tasks to explore universal computational principles.
The models allows for neural variability, but still assumes that all neurons are functionally similar. Could this be a problem when looking at other brain regions? Specially if the proportion of excitatory cells is smaller.


**Limitations:**

The authors mention some limitations and future work in their paper. However, it would be important to acknowledge the computational costs, training times, and data demands associated with the proposed method.

---

> ### Author Rebuttal · Authors · 2023-08-10
>
> Thank you for your thoughtful review of the paper! We are very excited to hear that you found the paper to be “presented clearly and is technically sound” and also agree that “extracting shared variability across multiple datasets is crucial for neuroscience”.
>
> In what follows, we will provide point-by-point replies to your questions.
>
> > 1. “it should be noted that this strategy heavily relies on having access to multiple large neural recordings, which may not be feasible for most basic neuroscience research where datasets are often limited.”
>
> **Reply:** We want to clarify two points regarding POYO’s ability to help in settings where data is limited. First, we have a single-session variant of our model (POYO-[single session]) which is not pre-trained on any data but is randomly initialized and trained end-to-end on a single session worth of data. This model, even without access to additional recordings for pretraining, outperforms other single-session methods (Table 2). In addition to working well on single sessions, the method is very sample efficient: In Figure 4, we show that our model can be trained from scratch from as little as 8 trials worth of data (which is less than 2 minutes), and is more data-efficient than a GRU model.
>
> Second, we agree that neural datasets are usually limited; this is a core motivation for our work and the creation of a large-scale pretrained model. This work shows that it is possible to combine multiple heterogeneous datasets that are collected in different labs, for different behavioral tasks, and under different experimental conditions, to provide a pretrained model that can transfer well on new sessions (Figure 3). Hence, we advocate for leveraging such an approach to make best out of available datasets. For example, a smaller lab with limited access to data could take a POYO model trained on much larger, open datasets (such as those available on DANDI) and then fine-tune it on their smaller dataset. We believe that expanding our method to decode various behavioral signals within smaller datasets will be an important objective for future work in order to demonstrate this potential use of POYO.
>
> > 2. “I would be interested in applying this method to even more diverse recordings throughout the brain and tasks to explore universal computational principles. The model allows for neural variability, but still assumes that all neurons are functionally similar.”
>
> **Reply:** The question of across region generalization and applying the model to larger-scale across-region questions is an exciting open area for future work! Even now in POYO-1, there is a lot of diversity in the location and areas for units in the different recordings, which span M1, PMd, and S1 (see Figure 1 in the attached pdf). These regions are long known to have distinct firing properties, cell types, and underlying dynamics. Yet, we are able to capture and exploit the activity of these diverse cells in a common model. We think POYO will be an extremely valuable tool in future work to better understand commonalities and differences in the neural code in different regions, both by testing generalization across regions and by studying the properties of models jointly trained on many regions.
>
> > 3. Could this be a problem when looking at other brain regions? Especially if the proportion of excitatory cells is smaller.
>
> **Reply:** In preliminary experiments, we have started to test the model on datasets from the visual cortex with good success. We plan to expand the model and training datasets in a next publication. In principle, we don’t believe that the number of excitatory cells will be a limiting factor, as long as the model is pre-trained on a diverse set of brain regions with different numbers of excitatory cells.
>
>
> > 4. The authors mention some limitations and future work in their paper. However, it would be important to acknowledge the computational costs, training times, and data demands associated with the proposed method.
>
> **Reply:**  Thank you for bringing this up. We discuss some of the computational requirements of our method in Appendix A.4.1, but we plan to include a paragraph in the discussion on this topic. Currently, we show that our largest model, trained on billions of tokens, uses 8 GPUs over a few days. As we increase the number of datasets and the complexity of the model, we expect that computational requirements will increase, but we are optimistic about recent advances in model scaling that have been enabled by breakthroughs in NLP. In terms of data demands, we have shown that large recordings are not a requirement to achieve performance that is competitive with existing methods, but our results suggest that we get significant boosts when a lot of data is leveraged. In settings where a lot of data has not been collected yet, we think that an important question for future work is whether we can leverage learned patterns of the neural code across various brain regions.
>
> That being said, we see the ability to train on a large-scale model, and use it in new settings via. unit identification or finetuning to be the key objective here, as the large model can be trained once but leveraged many more times. To give a concrete example, we plan to publicly release POYO-1 which can be tuned by anyone that has recordings from motor areas of non-human primates. Since the model is pre-trained, it can be tuned with a few samples (sample-efficient, Fig 4), it converges in a small number of steps (training time, Fig 4), it can be done with a single GPU or even just a CPU in less than an hour (Appendix A.4.1), and it does not require extensive hyperparameter tuning (all our transfer results in Table 2 were obtained using the same hyperparameters despite major differences across datasets), making our model accessible. We will highlight these advantages in the manuscript.

---

> > ### Comment · Reviewer_BY6C · 2023-08-15
> >
> > Thank you for your detailed and thorough responses! I still believe that this is a worthwhile contribution to the NeurIPS community.

---

### Official Review · Reviewer_Vmcn · 2023-07-06

**Soundness:** 4 excellent
**Presentation:** 4 excellent
**Contribution:** 4 excellent
**Rating:** 8
**Confidence:** 4

**Summary:**

This paper introduces a novel method for developing models that can predict the activity of neural populations by learning from data recorded across different sessions, tasks, and animals. The method is uses a custom tokenization procedure for spikes and a deep neural network architecture based on PerceiverIO. The paper presents the core method, then describes a technique to reuse a large pre-trained model to learn efficiently to predict a new neuronal population/condition, and finally presents a series of validation experiments.



**Strengths:**

- the proposed technique is technically solid, elegant, and performs well in the experiments.

- this technique addresses a real need in neuroscience labs, making the paper potentially very high impact.

- I found the scaling analysis (lines 237-257) particularly instructive, and especially relevant because for a new method like this one the reader may wonder what type of performance they may expect on their own data, as a function of the architectural choices they would have to take when deploying the method. This is of course a very hard thing to assess correctly, but this type of analysis is a great starting point.

- the paper is very well written.


**Weaknesses:**

- I have noticed a few minor issues (mostly related to clarity and plots) in section 3.4. See below under "questions".

- The discussion of related literature seems to imply that all spike train analysis has always been done by binning spikes. It would be good to include at least a cursory mention of kernel methods for spikes and other binless measures: see for instance Paiva, A.R.C., Park, I., Príncipe, J.C., 2010. A comparison of binless spike train measures. Neural Comput & Applic 19, 405–419.


**Questions:**

- line 277: "we see a bit of a dip in overall accuracy": how much is a bit? is this data shown anywhere?

- in figure 4, the blue/green colors are hard to see in grayscale. Can you choose colors that render better in grayscale?

- in figure 4, the compute efficiency plots don't seem to be cited anywhere in the text?

- line 289: "is competitive with other baselines". What other baselines?

- in the paragraph "Performance on new animals performing new tasks", it is not clear how much data was using for training and how much for testing. Can you clarify?


**Limitations:**

Limitation of the present work are appropriately discussed. I see no potential issue with societal impact.

---

> ### Author Rebuttal · Authors · 2023-08-10
>
> Thank you for your thoughtful review of the paper! We are very excited to hear that you found the proposed technique to be “technically solid and elegant” and also agree that it “addresses a real need in neuroscience labs”.
>
> In what follows, we will provide point-by-point replies to your questions.
>
> > 1. “The discussion of related literature seems to imply that all spike train analysis has always been done by binning spikes. It would be good to include at least a cursory mention of kernel methods for spikes and other binless measures”
>
> **Reply:**  Thanks for the suggestion! We will update the related work to include more context and mention other binless methods, such as the paper you cited and related work on kernel methods. When contrasting our spike-based tokenization scheme with other binning-based approaches, we were referring specifically to the deep learning for neural data literature here, where binning has been the approach to-date, because binned representations regularize the structure of spiking data and enable compatibility with existing machine learning frameworks, like MLPs, RNNs, and Transformers.
>
> > 2. “Line 277: "we see a bit of a dip in overall accuracy": how much is a bit? Is this data shown anywhere?”
>
> **Reply:**  This is a comment on the results from Table 2. More specifically, the two columns corresponding to Monkey T performing CO and RT tasks respectively. We use POYO-MP, which is pre-trained on multiple sessions from Monkey C and M, and transfer it to the new sessions from Monkey T, using unit identification (meaning that the transformer weights are frozen). We show that despite tuning a small number of parameters (unit embeddings and session embedding), we achieve an accuracy that is only 1% away from the accuracy we obtain when we train all weights from scratch, the latter setting corresponding to POYO-[Single-session]. This is the “dip” we are referring to. We will update the description to be more specific.
>
> > 3.  In figure 4, the blue/green colors are hard to see in grayscale. Can you choose colors that render better in grayscale?
>
> **Reply:** Yes, we will update the colors to work well in grayscale in a revision.
>
> > 4. In figure 4, the compute efficiency plots don't seem to be cited anywhere in the text?
>
> **Reply:** Thank you for pointing this out. We have revised the text to include a reference to Figure 4 and will include further discussion on the significance of this result in a final version of the paper.
>
> > 5.  Line 289: "is competitive with other baselines". What other baselines?
>
> **Reply:** We are referring to the baselines in Table 2, including the Wiener Filter, MLP, and AutoLFADS. We will make sure to revise this section to make it more clear.
>
> > 6. In the paragraph "Performance on new animals performing new tasks", it is not clear how much data was used for training and how much for testing. Can you clarify?
>
> **Reply:** We use the splits defined in the NLB benchmark. For NLB Maze, there are 1721 training samples and 574 testing samples. For NLB RTT, there are 810 training samples and 270 testing samples.

---

> > ### Comment · Reviewer_Vmcn · 2023-08-15
> >
> > Thank you for addressing the points I raised! The additional work carried out in reply to the other reviewers further strengthens this paper. I confirm my score.

---

### Official Review · Reviewer_c1Lq · 2023-07-07

**Soundness:** 3 good
**Presentation:** 2 fair
**Contribution:** 3 good
**Rating:** 6
**Confidence:** 4

**Summary:**

The authors describe a novel method for the task of neural decoding: using the time series of activity recorded from a population of neurons to predict the activity of scientifically relevant target variables. The describe their approach, called POYO, based upon the tokenization of spike data, the application of transformer based models to this data to generate pre-trained models, and the use of these pre-trained models for neural decoding. They demonstrate that this approach allows for accurate generalization across different sessions of neural recordings from the same animal, and across different animals.

**Strengths:**

The authors present a novel approach for the decoding of neural data, that appears promising in its ability to generalize over different animals and experimental sessions, which is a major problem in neuroscientific data analysis. The performance of their method is impressive, especially in the setting of pretrained models that are finetuned to novel datasets. The ability to combine various datasets in the way described here has the potential to be impactful.

**Weaknesses:**

My main concerns are with regard to scientific rigor, and the scope of the claims made in this work.
- Generalization across brain regions. I believe there needs to be a more thorough discussion of the role of brain regions in tasks where POYO is asked to generalize across individuals with recordings from different brain areas. To what degree will region-specific neural activity impact the ability of POYO to generalize when performing few shot learning? Concretely, what would behavioral decoding results in Table 2 look like if they are broken out per individual brain areas?
- Lines 37-39: "Overall, this lack of correspondence across recordings and channels complicates the integration of information from different experiments and individuals, ultimately hampering efforts to construct a unified perspective on population-level interactions and dynamics in the brain."It is not clear to me how POYO will address these challenges. While it is clear that POYO allows for highly performant decoding, the lack of recurrent structure within POYO makes it difficult to understand how this model would assist these challenges. Perhaps analysis of the attention mechanism could provide some answers to this question, but that point is not discussed in this work.
- Comparison with existing baselines. In Table 1, it is unclear to me why the performance of AutoLFADS + Linear only provided for the NLB datasets. It would be useful to see the performance of this model on all datasets, as it is a more powerful and commonly used decoding model than the others provided. Comparison with other transformer based neural decoding approaches like NDT (Ye and Pandarinath 2021) would also provide a better perspective on the value of the methodological advances proposed in this particular work.

I am open to reconsidering my score if my concerns here are addressed.

**Questions:**

- How many steps/epochs are fine tuning and unit identification run for? I was unable to find this information in the appendix.
- Line 55: the phrase "neural scaling laws" is ambiguous. Can it be changed to "scaling laws"?
- Line 349-350: "AutoLFADS extends the LFADS framework with Population Based Training to perform hyperparameter tuning, but has not been demonstrated for multi-session alignment."Is there any reason to believe that AutoLFADS is not capable of performing multi-session alignment? To my knowledge, Auto-LFADS is a optimization routine for LFADS. I would assume that it works for multi-session recordings as well.


**Limitations:**

As noted by the authors, generalizing this framework across other brain regions (as well as many other conditions) is a key limitation. In general, I have reservations about the use of the phrase "foundation model" without qualification in the context of neural decoding. Is the suggestion of this work that neuroscientists should aim to build a general purpose model that performs decoding regardless of model organism, brain area, or recording modality? While the results of this study are impressive, I do not believe that such claims are justified, even as future work.

---

> ### Author Rebuttal · Authors · 2023-08-10
>
> Thank you for your comments and questions, and for finding that our work "has the potential to be impactful"! In what follows, you will find our point-by-point response to your main concerns, and results from new experiments that we ran to address your feedback.
> Please let us know if there’s anything we can clarify further.
>
> > 1. “I believe there needs to be a more thorough discussion of the role of brain regions in tasks where POYO is asked to generalize across individuals with recordings from different brain areas.”
>
> **Reply:** Thank you for your suggestion. In the final paper, we plan to add a detailed discussion of the functional roles of the areas studied, and provide a new visualization of the recording sites across all the datasets that we study (see Figure 2 in attached PDF). Our current efforts have focused on neural recordings in three regions—M1, PMd, S1—that are intimately involved in planning and executing movements. While their functional role in movement and sensation are unique, many studies including ours have shown that they can all provide information that is used to guide decoding of movement. Yet, while different regions serve specific functions during voluntary movement (e.g., contracting muscles and sensing the state of the limb), they may perform these functions using common principles in the neural code.
>
> > 2. “To what degree will region-specific neural activity impact the ability of POYO to generalize when performing few shot learning?”
>
> **Reply:**  The datasets we consider in this paper broadly sample reach-related activity across PMd, M1 and S1. These three regions are highly distinct both functionally and based on cytoarchitecture and anatomy. For example, M1 and S1 have been shown to have highly distinct dynamics underlying their population activity. Furthermore, even within a region such as M1, our recording sites are distributed across varying functional sub-regions (Figure 2) and represent a great deal of diversity in terms of the underlying neural functions and firing patterns measured.
>
> In terms of across-region transfer of the model, we believe it is an important question that requires very thorough investigation, and a careful selection of pre-training and fine-tuning datasets which is outside of the scope of this work. We will make sure to include many of these points in our discussion.
>
> > 3. “Lines 37-39: "[...] The lack of recurrent structure within POYO makes it difficult to understand how this model would assist these challenges.”
>
> **Reply:** While we have not yet tackled this question, as the reviewer notes, analysis of the attention heads provides one possible way of using POYO to understand population-level interactions. We will add some discussion of this possible use to the manuscript.
> With regards to the comment that there is a “lack of recurrent structure within POYO”, the transformer architecture that we leverage provide powerful and general building blocks for modeling complex sequential data and time-series. Perhaps we misunderstood your point. If you can clarify, we will do our best to address your concern!
>
> > 4. “In Table 1, it is unclear to me why the performance of AutoLFADS + Linear only provided for the NLB datasets.”
>
> **Reply:** AutoLFADS is one of the main baselines in the NLB benchmark and the authors provide code and hyperparameters for the NLB datasets. Thus, we were able to reproduce these numbers. However, we were not able to obtain good results without further hyperparameter tuning on the other datasets in Table 2.
>
> Since the original submission, we have been able to run our datasets on NeuroCAAS, which is a cloud service containing the AutoLFADS tuning procedure, to obtain results for 4 of the datasets below.
>
> | | Monkey C, CO 10/13 | Monkey C, CO 10/21 | Monkey T, RT 08/20 | Monkey T, RT 09/06 |
> | - | - | - | - | - |
> | AutoLFADS + Linear | 0.9292 | 0.9519 | 0.4116 | 0.5061 |
> | POYO | 0.9603 | 0.9759 | 0.7986 | 0.8306 |
>
> As can be seen, POYO outperforms AutoLFADS on all of the datasets tested, with large gaps on RT. We will continue these runs as they are compute intensive, and we have 14 additional datasets that we report results on in Table 2.
>
> > 5. Comparison with other transformer based neural decoding approaches like NDT would also provide a better perspective on the value of the methodological advances proposed in this particular work.
>
> **Reply:** Thank you for this suggestion. We were able to reproduce NDT on both NLB datasets, and run other transformer models (see Table 1 in the General Response). We extensively tuned the hyperparameters of all baselines to ensure a fair comparison. Our results suggest that POYO (both single-session or pretrained) outperforms other transformer approaches.
>
> ** We would also like to note that we attempted to train NDT on the rest of the datasets, but we faced a similar challenge with hyperparameter tuning. We will continue to tune these baselines and hope to include them on all the evaluation data in Table 2.
>
> > 6. How many steps/epochs are fine tuning and unit identification run for?
>
> **Reply:** Fine tuning and unit identification are run for 50 epochs, with a batch size of 128. We have updated the appendix to include these details.
>
> > 7. “Is there any reason to believe that AutoLFADS is not capable of performing multi-session alignment? To my knowledge, Auto-LFADS is a optimization routine for LFADS.”
>
> **Reply:** Auto-LFADS has not been demonstrated in the multi-session condition and code for multi-session HPO has not been released. While, in theory, AutoLFADS can be applied to the multi-session stitching condition, it is unclear how to tackle the initialization of the alignment matrix for each session, or what HPO strategy to use with the presence of multiple datasets.
> We also note that multi-session stitching has only been done in sessions that are recorded from the same subject in the same brain region (and same chronic implant) during a stereotyped center-out task.

---

> > ### Comment · Reviewer_c1Lq · 2023-08-14
> > **Thank you.**
> >
> > I thank the authors for their thoughtful response, which clearly demonstrates significant additional work.
> >
> > I appreciate the discussion of recording sites (points 1 and 2 in the rebuttal), as well as Figure 2 included with the rebuttal. My concern is less about the inherent diversity of the brain regions included, and moreso how the specific sampling of brain regions in each animal affects affected generalization performance. It would be important to reference the information in Figure 2 of the attached pdf with the results from section 3.4, especially regarding generalization to new animals. In particular, I find Figure 2 makes it easier to appreciate that a pretrained model which has not seen recordings from S1 performs well on S1 data (Lines 283-293).
> >
> > Likewise, I am happy to see the additional comparisons provided by the authors, especially regarding existing transformer based decoding models. I am now more strongly convinced of the specific innovations proposed in this work.
> >
> > I can be more clear about point 3 in the rebuttal. Models like LFADS fit an explicit recurrent model of neural dynamics corresponding to an observed sequence of neural data, and part of their value is that after fitting, one can study properties of the inferred dynamical system. In contrast, POYO's focus and demonstrated results are pure prediction. While these are complementary approaches to the general problem of neural decoding, it is not clear to me how POYO's approach can be mapped onto insights about neural dynamics.
> >
> > Finally, I remain concerned about the use of the phrase foundation model, as also mentioned by reviewer Mfux. I would support the suggestion of mentioning the topic in the Discussion, but not elsewhere in the text. I would also ask the authors to devote significantly more effort to clarify in the main text how others can access their POYO models and use them on their own data (i.e. acess to code repository, instructions for working with pre-trained models).
> >
> > I have updated my score to reflect the changes made in the rebuttal.

---

> > > ### Author Response · Authors · 2023-08-19
> > >
> > > Thank you for your response and for updating your score! We are happy that our responses and new experiments addressed your concerns.
> > >
> > > Re: point 3 and recurrence. Thank you for the clarification. We will add some discussion and comparisons of POYO with dynamics-based approaches in the related work, and will discuss future interpretability experiments needed to better resolve population-level interactions learned by the model. Thanks again for your questions and suggestions!
> > >
> > > Re: “foundation model”. We have removed the mention of “foundation model” in the Results section when describing our multi-lab model. We plan to spend more space in the discussion to unpack the implications of our work and talk about next steps that would be needed to incorporate a self-supervised objective into our multi-session architecture.
> > >
> > > Re: “Instructions for using our pretrained models". We are planning to make the code and the models public, and will provide clear instructions in the main text.

---

### Author Rebuttal · Authors · 2023-08-10

We would like to thank all of the reviewers for their great feedback and suggestions! The reviewers agreed on the impact of the work and acknowledged the innovations behind the work for multi-session neuroscience.

Some highlights and praise from the reviewers:
- **Method and Approach:** “the proposed technique is technically solid, elegant, and performs well in the experiments.” (Vmcn) “a novel way to tokenize neural activity” (BY6C) “Efficient representation of the recordings via event-based tokenization” (Mfux)
- **Impact:**  “The ability to combine various datasets in the way described here has the potential to be impactful.” (c1LQ) “this technique addresses a real need in neuroscience labs, making the paper potentially very high impact.”  (Vmcn) “a major problem in neuroscientific data analysis” (c1LQ) “Extracting shared variability across multiple datasets is crucial for neuroscience” (BY6C) “proposed strategy could lay the foundation for studying unifying principles of neural computation”
- **Scaling analysis and rigor of experiments:** “I found the scaling analysis (lines 237-257) particularly instructive” (Vmcn) “Use of multiple experiments done in different labs” (Mfux) “accuracy of the method seems to improve significantly upon strong baselines” (Mfux)
- **Writing and Presentation:** “the paper is very well written.” (Vmcn) “The paper is presented clearly and is technically sound.” (BY6C)

Based upon reviewer comments, we ran a number of new experiments, and are currently working on the following revisions to our original submission:
- **Additional Baselines and Evaluations (c1LQ):** In response to rev (c1LQ)’s comments about including further single-session baselines, we conducted a number of new experiments to compare with the requested baselines (AutoLFADS, NDT [1]), in addition to a supervised-variant of NDT and another supervised neural data transformer baseline, EIT [2]. Across the board, we find that POYO-[Single-session] outperforms the other transformer-based approaches that use binning-based tokenization of the neural activity. Finetuning POYO-MP provides even further improvements in performance beyond the single-session models in the RTT task, demonstrating the power of having more data to pretrain on.

| | NLB-Maze | NLB-RTT |
| - | - | - |
| NDT + Linear | 0.8929 | 0.5895 |
| NDT-Supervised | 0.8708 | 0.4621 |
| EIT | 0.8791 | 0.4691 |
| AutoLFADS + Linear | 0.9062 | 0.5931 |
| POYO-[Single-session] | 0.9470 | 0.6850 |
| POYO-MP | 0.9466 | 0.7318 |
_Table 1: Behavioral decoding performance on NLB Datasets._

- **Neuron shuffling and identification experiment (Mfux):**  In response to reviewer Mfux’s great suggestion to conduct a neuron shuffling experiment, we ran unit-identification on a session that was already seen during pre-training. We test whether finetuning on this “new” set of units, will map the “new” units close to their true embedding (found during pre-training). We use our largest model, POYO-1, start with randomly initialized unit embeddings and run unit-identification. To compare the newly calibrated set of units with its pre-trained version, we normalize the unit-embeddings and report the cosine similarity averaged over the set of units. Note that normalization is done because POYO’s first cross-attention layer applies layer norm to these embeddings. We also report the “unit identification accuracy” which we define as the ratio of tuned unit embeddings that have their true unit embedding as their nearest neighbor.

| Dataset | # of units | Unit-Identification Accuracy | Unit-Identification Cosine-Similarity |
| - | - | - | - |
| Monkey C, CO, 2013/10/03 | 73 | 1.000 | 0.845 |
| Monkey M, CO, 2014/02/03 | 116 | 0.966 | 0.802 |
_Table 2: Unit re-identification results._

We include, in Figure 1 of the accompanying pdf, a visualization of the unit embeddings over multiple training steps during the unit-identification stage. We show how the new unit embeddings, which are initially random, converge towards their true embeddings, determined during pre-training.

Overall, this analysis suggests that the unit-identification approach is robust and reliable for identifying units. We believe that understanding the functional similarities of units that are mapped close to each other in the unit embedding space will be an interesting avenue for future work.

**Impact of the work:** The proposed framework achieves a number of new firsts. POYO is the first to integrate and decode population activity across multiple animals using a unified, common model, and is also the first to jointly integrate datasets from different labs and behavioral tasks. The diversity of datasets that we have demonstrated our approach on is staggering: we have data from 9 nonhuman primates and over 150 sessions, train on diverse datasets from different spike sorting algorithms and threshold crossings, and across diverse behavioral tasks from 4 different research labs that are measured and executed through completely different manipulandum and at different sampling rates.

---

References:

[1] Ye et al., "Representation learning for neural population activity with Neural Data Transformers." Neurons, Behavior, Data analysis, and Theory 2021.

[2] Liu et al., “Seeing the forest and the tree: Building representations of both individual and collective dynamics with transformers” NeurIPS 2022

---

### Author Response · Authors · 2023-08-19

We thank all the reviewers for their responses and for engaging in the discussion! We believe that our work is now stronger thanks to all of the reviewer feedback. We plan to use the additional page to include our additional experiments and elaborate more on the discussion points brought up during the review.

---

### Decision · Program_Chairs · 2023-09-21

**Decision:**

Accept (poster)

**Comment:**

This paper received unanimous support from all the reviewers. There is general agreement that the problem targeted is important and that the performance of the neural decoding algorithm is impressive. There were initial concerns regarding some of the claims and the experimental evaluation that appeared to have been well addressed by the rebuttal. The AC thus recommends the paper to be accepted.